# Infectiousness of Asymptomatic *Meriones shawi*, Reservoir Host of *Leishmania major*

**DOI:** 10.3390/pathogens12040614

**Published:** 2023-04-18

**Authors:** Jovana Sadlova, Barbora Vojtkova, Tereza Lestinova, Tomas Becvar, Daniel Frynta, Kamal Eddine Benallal, Nalia Mekarnia, Zoubir Harrat, Petr Volf

**Affiliations:** 1Department of Parasitology, Faculty of Science, Charles University, Vinicna 7, 128 44 Prague, Czech Republic; vojtkova.bara@gmail.com (B.V.); terka.kratochvilova@seznam.cz (T.L.); tomas.becvar@natur.cuni.cz (T.B.); benallalkamel4@yahoo.fr (K.E.B.); volf@cesnet.cz (P.V.); 2Department of Zoology, Faculty of Science, Charles University, Vinicna 7, 128 44 Prague, Czech Republic; frynta@centrum.cz; 3Laboratory of Eco Epidemiology of Parasitic Diseases and Population Genetics, Institut Pasteur d’Algérie, 16045 Algiers, Algeria; nalia.mekarnia@univ-reims.fr (N.M.); zharrat@gmail.com (Z.H.); 4EA ESCAPE 7510, Faculty of Pharmacy, Université de Reims Champagne-Ardenne, 51 Rue Cognacq-Jay, 51096 Reims CEDEX, France

**Keywords:** *Leishmania*, *Phlebotomus*, *Meriones*, reservoir host, asymptomatic infection, xenodiagnosis

## Abstract

Leishmaniases are neglected diseases caused by protozoans of the genus *Leishmania* that threaten millions of people worldwide. Cutaneous leishmaniasis (CL) caused by *L. major* is a typical zoonosis transmitted by phlebotomine sand flies and maintained in rodent reservoirs. The female sand fly was assumed to become infected by feeding on the skin lesion of the host, and the relative contribution of asymptomatic individuals to disease transmission was unknown. In this study, we infected 32 *Meriones shawi*, North African reservoirs, with a natural dose of *L. major* obtained from the gut of infected sand flies. Skin manifestations appeared in 90% of the animals, and xenodiagnosis with the proven vector *Phlebotomus papatasi* showed transmissibility in 67% of the rodents, and 45% were repeatedly infectious to sand flies. Notably, the analysis of 113 xenodiagnostic trials with 2189 sand flies showed no significant difference in the transmissibility of animals in the asymptomatic and symptomatic periods; asymptomatic animals were infectious several weeks before the appearance of skin lesions and several months after their healing. These results clearly confirm that skin lesions are not a prerequisite for vector infection in CL and that asymptomatic animals are an essential source of *L. major* infection. These data are important for modeling the epidemiology of CL caused by *L. major*.

## 1. Introduction

Leishmaniases are diseases caused by parasitic protozoans of the genus *Leishmania* (Kinetoplastida: Trypanosomatidae) endemic in 98 countries of 5 continents, where they threaten more than 1 billion people. There are three main forms of leishmaniasis: the most severe form is visceral leishmaniasis, which can be fatal without treatment. Mucocutaneous leishmaniasis is a disabling form producing mucosal lesions and cartilage destruction in the oronasal and pharyngeal regions. The most common form is cutaneous leishmaniasis (CL). Skin lesions heal spontaneously in most patients but can leave stigmatic scars for life. It is estimated that 600,000 to 1 million new cases of CL occur worldwide annually but only around 200,000 are reported to the WHO [1].

Cutaneous leishmaniasis caused by *L. major* is a typical zoonosis maintained in rodents. The main reservoir hosts have been shown to be the Great Gerbil *Rhombomys opimus* and jirds of the genus *Meriones* in Central and SW Asia [2]; the Fat Sand Rat *Psammomys obesus* and *Meriones* species in North Africa and the Middle East [3] and species of the genera *Arvicanthis* and *Mastomys* in Sub-Saharan Africa [4,5]. Other species of rodents and other mammals naturally infected with *L. major* may also be involved in local transmission cycles, but their role has yet to be confirmed.

Field reports had revealed a high representation of asymptomatic individuals among rodent reservoirs infected with *L. major* [6,7,8,9]. The importance of asymptomatic hosts in the epidemiology of leishmaniasis has so far been studied and discussed in only visceral leishmaniasis caused by *L. donovani* and *L. infantum* [10,11], and mathematical modeling has suggested the importance of asymptomatic infection in the transmission of *L. donovani* [12]. Therefore, asymptomatic hosts could pose a major challenge to control programs if their infectiousness is confirmed [10]. Indeed, a meta-analysis of xenodiagnostic studies by Quinnell and Courtenay (2009) [13] confirmed that the infectiousness of dogs infected with *L. infantum* increased with the clinical severity of the disease, but since the infectiousness of asymptomatic dogs was sufficiently high, control must target both asymptomatic and symptomatic dogs.

However, the comparative contributions of asymptomatic and symptomatic hosts to the transmission of CL remain unknown. Therefore, we have established the laboratory colony of *Meriones shawi*, one of the main reservoir species in North Africa. The colony was derived from animals captured in M’Sila, the Algerian province most affected by CL [14]. Using a *Leishmania* isolate from the same locality and a proven insect vector in the area, *Phlebotomus papatasi*, we obtained a natural model for studying important factors affecting the outward transmission of *L. major*, including the infectiousness of asymptomatic animals.

## 2. Materials and Methods

### 2.1. Sand Flies, Parasites and Rodents

The colony of *P. papatasi* originating from Turkey has been maintained in the insectary of the Department of Parasitology, Charles University in Prague, under standard conditions (26 °C on 50% sucrose, humidity in the insectary 60–70% and 14 h light/10 h dark photoperiod) as described previously [15].

The human isolate *L. major* MHOM/DZ/2009/LIPA100/MON-25 from M’Sila region in Algeria was used. *Leishmania* were fluorescence-marked with m-Scarlet according to the methodology described by [16] to enable analysis of the distribution of parasites on a microscale (these analyses will be published separately). Promastigotes were cultured in M199 medium (Sigma-Aldrich, Merck, Darmstadt, Germany) containing 10% heat-inactivated fetal bovine calf serum (FBS, Gibco, Thermo Fisher Scientific, Waltham, MA, USA) supplemented with 1% BME vitamins (Basal Medium Eagle, Sigma-Aldrich, Merck, Darmstadt, Germany), 2% sterile human urine and 250 μg/mL amikacin (Amikin, Bristol-Myers Squibb, New York, NY, USA). 

The breeding colony of *M. shawi* was established in the LEEPGP animal facility at the Pasteur Institut of Algiers from animals originating in M’Sila, Algeria, and then bred at the animal facility of the Faculty of Science, Charles University in Prague. Animals were maintained in T4 breeding containers (Velaz, Prague, Czech Republic) equipped with bedding (Krmne smesi Kvidera, Spálené Poříčí, Czech Republic), breeding material (Woodwool, Ratiboř, Czech Republic) and hay (Krmne smesi Kvidera, Spálené Poříčí, Czech Republic), provided with a standard feed mixture ST-1 (SubliCZ.cz, Sojovice, Czech Republic) and water ad libitum, with a 12 h light/12 h dark photoperiod, temperature 22–25 °C and humidity 40–60%.

### 2.2. Experimental Infections of Sand Flies

*Phlebotomus papatasi* females were infected with *L. major* according to [17]. Briefly, parasites from log-phase cultures were resuspended in heat-inactivated defibrinated ram blood at a concentration of 2 × 10^6^ promastigotes/mL. Sand flies were allowed to ingest this infectious suspension through a chick-skin membrane and after feeding were maintained in the same conditions as the colony. For experimental infections of rodents, female sand flies were dissected 8 days post-bloodmeal when mature infections had developed and metacyclic forms had accumulated in the thoracic midguts.

### 2.3. Experimental Infections of M. shawi

Rodents (at age 6–10 weeks) were anesthetized with a mixture of 50 mg/kg ketamine and 5 mg/kg xylazine and infected according to [18]. Parasites derived from thoracic midguts of infected sand flies were pooled in sterile saline and counted using a Bürker chamber. Dissected salivary glands (SG) of *P. papatasi* females were pooled (10 glands per 10 µL of saline) and stored at −20 °C. Before inoculating mice, SG were frozen and thawed three times in liquid nitrogen, and the resulting lysate was mixed into the inoculum. Finally, 5.5 µL of the mixture was intradermally injected into the left pinnae. The infection dose was either 7000 or 14,000 parasites corresponding to 1 or 2 infected sand fly guts with an addition of 0.5 SG. Animals were checked weekly for external signs of the disease. They were subjected to xenodiagnosis at 4–5-week intervals and sacrificed at various time intervals between weeks 15 and 38.

### 2.4. Xenodiagnoses

Rodents were anesthetized with ketamine/xylazine, and 30–40 *P. papatasi* females kept in small plastic tubes covered with fine nylon mesh were allowed to feed on pinnae of inoculated ears (Figure 1). On days 7–10 after feeding, the females were dissected, the guts were inspected by light microscopy, and the intensity and location of infections were assessed as previously described [17].

### 2.5. Tissue Sampling and Leishmania Detection Post-Mortem by qPCR

Rodents were sacrificed by cervical dislocation under anesthesia. Whole ears (inoculated and contralateral), whole ear-draining lymph nodes, whole spleen, liver (whole liver was homogenized and a quarter of the volume was used for analysis), paws (whole forepaws and skin and soft tissue from the bottoms of the hindpaws) and tail skin were removed and stored at −20 °C. For qPCR, extraction of total DNA from rodent tissues was performed using a DNA tissue isolation kit (Roche Diagnostics, Indianapolis, IN, USA) according to the manufacturer’s instructions. Parasite quantification by quantitative PCR (qPCR) was performed in a Bio-Rad iCycler & iQ Real-Time PCR Systems using the SYBR Green detection method (SsoAdvanced™ Universal SYBR^®^ Green Supermix, Bio-Rad, Hercules, CA, USA). Primers targeting 116 bp long kinetoplast minicircle DNA sequence (forward primer (13A): 5′-GTGGGGGAGGGGCGTTCT-3′ and reverse primer (13B): 5′-ATTTTACACCAACCCCCAGTT-3′) were used [19]. One microliter of DNA was used per individual reaction. PCR amplifications were performed in triplicates using the following conditions: 3 min at 98 °C followed by 40 repetitive cycles: 10 s at 98 °C and 25 s at 61 °C. Tissue from a non-infected animal was used as a negative control. A series of 10-fold dilutions of *L. major* promastigote DNA, ranging from 5  ×  10^3^ to 5  ×  10^−2^ parasites per PCR reaction was used to prepare a standard curve. Quantitative results were expressed by interpolation with a standard curve. To monitor non-specific products or primer dimers, a melting analysis was performed from 70 to 95 °C at the end of each run, with a slope of 0.5 °C/c, and 6 s at each temperature.

### 2.6. Statistical Analysis

To analyze the effects of symptoms and infection dose on the result of xenodiagnostic trials, we employed a marginal model accounting for the dependence of data collected from the same animal [20]. We set the actual presence/absence of symptoms and infection dose as fixed factors, animal identity as ID, binomial distribution, logit link function and ar1 correlation structure. Then, we ran a geeglm function as implemented in geepack (Generalized Estimating Equation Package; [21]). The calculations were performed in an R-environment (R Core Team, Vienna, Austria, 2021).

### 2.7. Animal Experimentation Guidelines

Animals were kept in the animal facility of Charles University in Prague in accordance with institutional guidelines and Czech legislation (Act No. 246/1992 and 359/2012 coll. on Protection of Animals against Cruelty in present statutes at large), which complies with all relevant European Union and international guidelines for experimental animals. All experiments were approved by the Committee on the Ethics of Laboratory Experiments of Charles University in Prague and were carried out based on permission no. MSMT-31778/2019-6 of the Ministry of Education, Youth and Sports. The investigators are certified for animal experiments by the Ministry of Agriculture of the Czech Republic.

## 3. Results

In total, 32 animals were infected; 12 received an infectious dose of 14,000 promastigotes (derived from 2 sand fly guts) and 20 were infected with 7000 promastigotes (derived from 1 sand fly gut). Skin manifestations developed in 29 animals; the first skin changes usually appeared 9–15 weeks after infection, regardless of the infective dose, first as a swelling or non-ulcerative nodule or directly as an ulcerative skin lesion (Figure 2a,b). In most animals, the lesions then grew and persisted until the end of experiments, but the skin symptoms healed into the asymptomatic state in 2 animals (animals No. 3 and 28) and ulcerative lesions healed to a non-ulcerative swelling in 3 animals (animals No. 22, 29 and 31). Out of 32 infected jirds, two individuals were asymptomatic throughout the experiment; they were killed relatively early (15–16 weeks p.i.), and PCR confirmed the presence of parasites. One animal killed at 30 weeks p.i. was classified as uninfected because it never showed external signs of disease, was not infectious to sand flies and PCR analysis post-mortem was negative.

Xenodiagnostic experiments were conducted continuously at 4–5-week intervals throughout the experiment to capture both asymptomatic and symptomatic phases of the disease in individual animals (Figure 2a, Appendix A). Overall, 20 of 30 animals (67%) were infectious to sand flies on at least one trial, and 14 of them were infectious repeatedly (in 2–5 trials). Six animals were infectious even 4–6 weeks before the appearance of skin symptoms, and in animals that healed lesions during the experiment, we observed infectivity even 14 and 17 weeks after the disappearance of skin manifestations. Analysis of all 113 xenodiagnostic trials with 2.189 sand flies showed no significant difference in the transmissibility of animals in the asymptomatic and symptomatic periods (geeglm: df = 1, Chi-square = 2.842, *p* = 0.0918). Moreover, the model revealed no significant difference in infectiousness between animals inoculated with different infective doses (df = 1, Chi-square = 0.7286, *p* = 0.3933). 

At the end of the experiments, animals were sacrificed and their organs were analyzed by PCR. The intensity of infections in positive tissues was further quantified by qPCR. *Leishmania* were mainly localized in inoculated left ears and their draining lymph nodes, but also spread to other skin areas: right ears, paws and tail, and to a lesser extent, they were detected in the spleen and liver (Table 1). While the parasite load was very high in inoculated ears, mostly only small numbers of leishmania (less than 100) were detected in other parts of the body.

## 4. Discussion

*Meriones shawi* is a North African rodent species distributed from Morocco to Egypt, inhabiting clay and sandy deserts and dry steppes, and it is present often also in cereal crops and orchards. It is a colonial species living socially in underground burrows with nocturnal and diurnal activity [22]. Together with *Psammomys obesus*, these species play the main role in the transmission of *L. major* in North Africa [4,23]. Host competence of *M. shawi* for *L. major* was also confirmed by a laboratory study that demonstrated the infectiousness of the animals to the vectors five months after subcutaneous infection of ears with 10^6^ promastigotes [24]. A thorough field study from central Tunisia, where both species live in sympatry, showed that the prevalence of the parasite is very high in both *M. shawi* and *P. obesus* (41% and 34%, respectively). Asymptomatic infections were observed in 40% of infected animals of both species, and among the symptomatic ones, skin lesions were localized mostly on the pinnae. The lesions appeared to be more destructive in *M. shawi* than in *P. obesus* [6]. In another study from Tunisia, only 35% of infected *P. obesus* had no visible ear lesions, but the ears may have been destroyed by another cause such as fighting, as 40% of those with ear lesions were negative for *L. major* when tested by footpad inoculation [7].

In reservoir hosts of *Leishmania*, it is generally assumed that the course of the disease is rather mild because it is also in the parasite’s interest to survive in these hosts for long periods of winter or rainy seasons when no vectors are present [25]. Thus, an equilibrium between the reservoir hosts and their parasites has necessarily been established that allows both organisms to survive long enough. In this respect, an excellent coexistence has been observed between *L. major* and its main host in Central Asia and neighboring countries, the Great gerbil, *Rhombomys opimus*. Laboratory infections showed that although these animals were susceptible, ulceration and visceralization never developed [26]. In field-trapped animals in northeastern and central Iran, the proportion of asymptomatically infected *R. opimus* reached 90% [8,9,27]. However, it should be noted that co-infections of *L. major* and *L. turanica* are common in endemic areas of Central Asia (reviewed in [2]), which may affect the immune response and overall course of *Leishmania* infections of rodents in this area.

In the field studies, it was not possible to distinguish whether asymptomatic animals were more resistant to leishmania than symptomatic ones, or have been infected with a less virulent variant of the parasite, or are pre- or post-symptomatic [6,7]. Our laboratory study showed that almost all *M. shawi* developed skin symptoms; however, symptoms developed relatively late, and some individuals can heal the lesions. Resolution of skin symptoms was observed in one-sixth of the animals, but the proportion of healed animals would probably be even higher if the experiments were extended for additional months.

We mimicked as closely as possible the natural course of initiation of infection. Sand fly salivary gland content has been added as salivary proteins and peptides with antihemostatic and immunomodulatory properties significantly affect the establishment of the parasite in the host (reviewed by [28,29]). Promastigotes were taken from thoracic midguts of infected sand flies: female sand flies were dissected 8 days post-bloodmeal, when mature *L. major* infection developed with a predominance of metacyclic stages. Infection dose was kept low, corresponding to parasite numbers in one or two infected thoracic midguts (7000 or 14,000 parasites, respectively). Intradermal needle inoculation allowed us to control the infectious dose, which is not possible with direct infection by sand fly bite, where different females inoculate the host with *Leishmania* numbers varying by several orders of magnitude [30,31]. Kimblin et al. (2008) [30] compared *L. major* infections initiated with different infection doses and showed that a low infection dose leads to less pathology but higher parasite number in the chronic phase, so that the host acts as an effective long-term reservoir of infection. With high dose (10^6^) of parasites in inoculum, the growth of skin lesions would be unnaturally rapid as was described not only for *L major* but also for *L. amazonensis* in mice model [32,33]. 

In all infected animals used in our experiments, parasites were most abundant in the inoculated pinnae; in two-thirds of the individuals, they also spread to other areas of the body. Smaller numbers were detected in the collateral ear, the skin of the paws and tail, as well as in the lymph nodes, spleen and liver. This pattern of *L. major* dissemination with higher representation in the skin than in internal organs was also observed in other wild rodent species: *Arvicanthis niloticus*, *A. neumanni*, *Mastomys natalensis*, *Cricetulus griseus* and *Lagurus lagurus*, while extensive visceralization under the same experimental conditions was observed in *Phodopus sungorus*, with 71% of spleen samples infected [5,34].

Importantly, also, pre-symptomatic animals and animals with healed skin symptoms were infectious to the vector and statistically, based on the whole material of 113 xenodiagnostic trials, there was no significant difference between infectiousness in the symptomatic and asymptomatic periods. It is interesting to note that some individuals were not infectious for sand flies, although they had ulcerative lesions and a significant parasite load in the pinnae post-mortem. We attribute this phenomenon to the heterogeneous distribution of parasites in the pinnae, as has been reported for *L. donovani* in the skin of BALB/c mice [35]. Our next research goal will be to study the dissemination of *L. major* at the microscale in their natural host, *Meriones shawi*. 

Until now, the infectiousness of asymptomatic hosts has been studied almost exclusively in visceral leishmaniasis. Xenodiagnostic experiments showed the transmissibility of *L. donovani* from asymptomatic BALB/c mice [18] and *L. infantum* from different mammalian hosts [36,37,38,39,40,41]. The infectiousness of asymptomatic dogs infected with *L. infantum* has been studied most intensively, and meta-analysis of data published up to 2009 by Quinnell and Courtenay [13] showed that the proportion of infectious dogs increased significantly with clinical severity, but asymptomatic dogs were also infectious and infected a similar proportions of flies as sick dogs. Canine studies also demonstrated that some dogs are “super-spreaders” while others contribute little to transmission (15% to 44% of dogs were responsible for >80% of all sand fly infections) [42]. The situation was different for the natural hosts of *L. major* in this study. The majority (67%) of *Meriones shawi* were infectious, and 47% of the animals were infectious repeatedly in different time intervals post-infection. 

In CL, it was previously assumed that the sand fly female becomes infected by feeding on the lesion, while asymptomatic individuals were not considered as a source of infection [43]. Only a few studies have described the infectiousness of asymptomatic hosts with CL, such as sloths *Choloepus hoffmanni* infected with *L. braziliensis* [44] or black rats infected with *L. tropica* [45]. Our study based on the natural model *Meriones shawi*–*L. major*–*P. papatasi* demonstrated the regular transmissibility of asymptomatic infected animals long before the appearance of skin lesions and long after their healing. Using a significant number of xenodiagnostic experiments, we have shown that skin lesions are not a prerequisite for vector infection, and their presence is not a good diagnostic marker for the presence of the parasite. These data are important for modeling the epidemiology of CL by *L. major* and for control management. These results also raise the question of the extent to which asymptomatic incidental hosts, including infected humans, may be involved in the transmission of CL. Only xenodiagnostic experiments covering both symptomatic and asymptomatic phases of infection can answer this question. 

## Figures and Tables

**Figure 1 pathogens-12-00614-f001:**
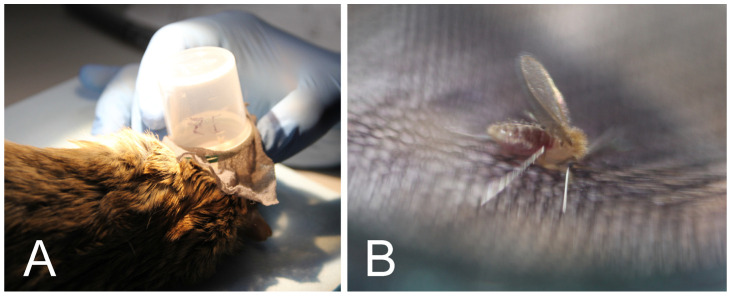
Xenodiagnosis. (**A**), Sand flies held in plastic tubes coated with nylon mesh on the pinna of *M. shawi*; (**B**), Detail of feeding female *P. papatasi*.

**Figure 2 pathogens-12-00614-f002:**
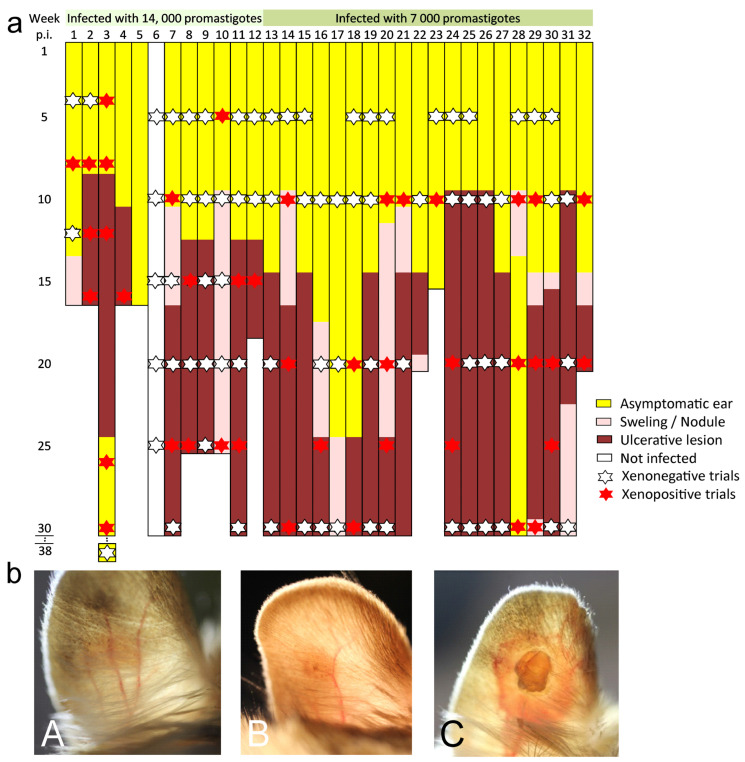
External manifestation and xenodiagnoses of *Leishmania major* in *Meriones shawi*. (**a**) Time course of external manifestations of infection on the pinnae of 32 animals. Xenodiagnostic experiments are labeled in the respective weeks after infection; positive trials are marked with a red asterisk, negative trials with a white asterisk. (**b**) Skin symptoms on pinnae. (**A**) Swelling; (**B**) Nodule; (**C**) Ulcerative lesion.

**Table 1 pathogens-12-00614-t001:** Distribution of *Leishmania major* and parasite load in organs and tissues of *Meriones shawi* detected by PCR and qPCR post-mortem.

Infective Dose	Week p.i.	Meriones No.	IE	CE	IE-LN	CE-LN	FP	HP	Tail	Spleen	Liver
14 × 10^3^	16	1	8.30 × 10^6^	0	<1 × 10^2^	0	0	0	0	<1 × 10^2^	<1 × 10^2^
14 × 10^3^	16	2	2.75 × 10^5^	0	0	0	0	0	<1 × 10^2^	0	0
14 × 10^3^	40	3 *	0	0	<1 × 10^2^	0	0	0	0	<1 × 10^2^	0
14 × 10^3^	16	4	3.43 × 10^6^	0	<1 × 10^2^	0	0	0	0	0	0
14 × 10^3^	16	5	2.13 × 10^4^	0	<1 × 10^2^	0	0	0	0	<1 × 10^2^	0
14 × 10^3^	30	6 **	0	0	0	0	0	0	0	0	0
14 × 10^3^	30	7	3.29 × 10^9^	<1 × 10^2^	0	0	0	<1 × 10^2^	<1 × 10^2^	0	0
14 × 10^3^	25	8	1.41 × 10^5^	0	0	0	0	0	0	0	0
14 × 10^3^	25	9	7.42 × 10^8^	<1 × 10^2^	0	0	<1 × 10^2^	<1 × 10^2^	<1 × 10^2^	0	<1 × 10^2^
14 × 10^3^	25	10	1.08 × 10^4^	0	0	0	0	0	0	0	0
14 × 10^3^	30	11	4.78 × 10^7^	<1 × 10^2^	0	0	0	<1 × 10^2^	<1 × 10^2^	0	0
14 × 10^3^	18	12	3.94 × 10^5^	0	<1 × 10^2^	0	<1 × 10^2^	0	0	0	0
7 × 10^3^	30	13	2.19 × 10^7^	<1 × 10^2^	0	0	0	<1 × 10^2^	<1 × 10^2^	0	<1 × 10^2^
7 × 10^3^	30	14	5.86 × 10^5^	<1 × 10^2^	0	0	0	0	<1 × 10^2^	<1 × 10^2^	<1 × 10^2^
7 × 10^3^	30	15	1.76 × 10^5^	0	<1 × 10^2^	0	0	0	<1 × 10^2^	0	0
7 × 10^3^	30	16	2.43 × 10^7^	0	0	0	0	0	0	0	0
7 × 10^3^	30	17	3.74 × 10^4^	0	0	0	<1 × 10^2^	0	0	0	0
7 × 10^3^	30	18	1.83 × 10^6^	0	0	0	0	0	0	0	0
7 × 10^3^	30	19	9.81 × 10^7^	0	0	0	0	0	0	0	0
7 × 10^3^	30	20	5.45 × 10^4^	0	0	0	0	0	0	0	0
7 × 10^3^	30	21	5.52 × 10^4^	0	0	0	<1 × 10^2^	0	<1 × 10^2^	<1 × 10^2^	0
7 × 10^3^	20	22	1.14 × 10^5^	<1 × 10^2^	1.75 × 10^6^	0	0	0	0	0	0
7 × 10^3^	15	23	2.66 × 10^6^	<1 × 10^2^	<1 × 10^2^	0	0	0	0	<1 × 10^2^	0
7 × 10^3^	30	24	4.68 × 10^6^	0	0	0	0	<1 × 10^2^	0	0	0
7 × 10^3^	30	25	1.67 × 10^8^	0	0	0	<1 × 10^2^	0	0	0	<1 × 10^2^
7 × 10^3^	30	26	1.32 × 10^5^	0	0	0	0	0	0	0	0
7 × 10^3^	30	27	7.46 × 10^3^	0	0	0	<1 × 10^2^	0	0	0	0
7 × 10^3^	30	28	1.56 × 10^3^	0	0	0	0	0	0	0	0
7 × 10^3^	30	29	3.35 × 10^5^	0	0	0	0	0	0	0	0
7 × 10^3^	30	30	3.46 × 10	0	0	0	0	<1 × 10^2^	1.14 × 10^3^	0	0
7 × 10^3^	30	31	7.06 × 10^2^	0	0	0	0	0	0	0	0

IE, inoculated ear; CE, contralateral ear; IE-LN, lymph node draining IE; CE-LN, lymph node draining CE; FP, forepaws; HP, hindpaws; *, healed ear infection; **, uninfected animal. Color differentiation of infection intensity: red, >10^8^; dark blue, 10^6^–10^8^, light blue, 10^4^–10^6^, green, 10^2^–10^4^, pink, <10^2^.

## Data Availability

Data are contained within the article or Appendix A.

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
