# Peer review of "Infectiousness of Asymptomatic Meriones shawi, Reservoir Host of Leishmania major"

_pathogens, 2023, doi:10.3390/pathogens12040614_

Round 1
Reviewer 1 Report
Thank you very much for this manuscript reporting the infectiousness of Meriones shawi based on the natural model reservoir-L. major-sand fly. The results are very interesting and highly relevant for the epidemiology of cutaneous leishmaniasis by L. major, highlighting the role of asymptomatic subjects in the parasite’s transmission dynamics. The manuscript is well written, and the methodology is appropriate. The results are very clearly presented.
I merely have a few remarks:
Introduction.
In the Introduction section (Lines 34-37), authors report general morbidity and mortality data of leishmaniases regardless of clinical form and etiology. I suggest the authors directly focus on cutaneous leishmaniasis. Moreover, recent data on prevalence and incidence may be cited.
Materials and methods.
-Line 106: Please write out the abbreviation PBM.
-Paragraph 2.5 "Tissue sampling and…." The authors evaluated the parasite load by qPCR reaction, so the result is not related to the sample weight or size (Table 1), unless DNA has been extracted from the whole organ. These data could provide additional information on parasite spread in tissues, allowing for a better comparison of the parasite load between tissues/organs. Furthermore, a figure showing the standard curve could be added as Supplementary material.
Results:
-Line 156. Unfortunately, I can understand why authors report 5 animals with lesions healed. In Figure 1, I see only 2 subjects that become asymptomatic by the end of the experiments. Moreover, the statement in Lines 156-157 is not a result, so I suggest moving it into the Discussion.
-Figure 1. Could the author enhance the quality of the bar referred to in subject n. 3 (30/38 weeks)?
Finally, despite having ulcerative lesions and a significant parasite load in their ear lobes postmortem, some cases (e.g., id 19 and 25) were not infectious at all.
Might the authors include a brief discussion on this topic?
I understand that this could be related to normal biological variation. Anyway, do the authors believe that this is due to an uneven distribution of parasites in tissues?
In other words, does the parasitic load differ at different sites on the earlobe (for example, is it higher in the section collected for qPCR?).
Best regards,
Author Response
Thank you very much for this manuscript reporting the infectiousness of Meriones shawi based on the natural model reservoir-L. major-sand fly. The results are very interesting and highly relevant for the epidemiology of cutaneous leishmaniasis by L. major, highlighting the role of asymptomatic subjects in the parasite’s transmission dynamics. The manuscript is well written, and the methodology is appropriate. The results are very clearly presented.
We sincerely thank the reviewer for the time spent on this review and the insightful comments. We greatly appreciate his positive evaluation of our work. Below are point-by-point responses to all comments. The changes in the text of the manuscript are marked in blue.
I merely have a few remarks:
Introduction.
In the Introduction section (Lines 34-37), authors report general morbidity and mortality data of leishmaniases regardless of clinical form and etiology. I suggest the authors directly focus on cutaneous leishmaniasis. Moreover, recent data on prevalence and incidence may be cited.
In the first paragraph of introduction, we have specified more details on different forms of leishmaniasis and data on prevalence is now given only for the cutaneous form. Instead of the older 2012 publication we cite the current WHO 2023 data:
“Leishmaniases are diseases caused by parasitic protozoans of the genus Leishmania (Kinetoplastida: Trypanosomatidae) endemic in 98 countries of 5 continents, where they threaten more than 1 billion people. There are three main forms of leishmaniasis: the most severe form is visceral leishmaniasis, which can be fatal without treatment. Mucocutaneous leishmaniasis is a disabling form producing mucosal lesions and cartilage destruction in the oronasal and pharyngeal regions. The most common form is cutaneous leishmaniasis (CL). Skin lesions heal spontaneously in most patients, but can leave stigmatic scars for life. It is estimated that 600 000 to 1 million new cases of CL occur worldwide annually but only around 200 000 are reported to WHO [1].”
Materials and methods.
-Line 106: Please write out the abbreviation PBM.
Done.
-Paragraph 2.5 "Tissue sampling and…." The authors evaluated the parasite load by qPCR reaction, so the result is not related to the sample weight or size (Table 1), unless DNA has been extracted from the whole organ. These data could provide additional information on parasite spread in tissues, allowing for a better comparison of the parasite load between tissues/organs. Furthermore, a figure showing the standard curve could be added as Supplementary material.
Thanks to the reviewer for this comment, we specified the sampling for DNA extraction in the methodology, paragraph 2.5. “Whole ears (inoculated and contralateral), whole ear-draining lymph nodes, whole spleen, liver (whole liver was homogenized and a quarter of the volume was used for analysis), paws (whole forepaws and skin and soft tissue from the bottoms of the hindpaws) and tail skin were removed and stored at −20 oC.”
The standard curve was generated individually for each run of qPCR analysis and the samples measured in that run were related to it. Since standard curves may change slightly between the runs, we did not add various standard curves because this would eventually impair the clarity of the data.
Results:
-Line 156. Unfortunately, I can understand why authors report 5 animals with lesions healed. In Figure 1, I see only 2 subjects that become asymptomatic by the end of the experiments. Moreover, the statement in Lines 156-157 is not a result, so I suggest moving it into the Discussion.
The 5 animals with signs of healing included three individuals in which ulceration disappeared and only swelling remained. We've specified it in the first paragraph of Results section and believe the text is now clear: “In most animals, the lesions then grew and persisted until the end of experiments, but the skin symptoms healed into the asymptomatic state in 2 animals (animals No. 3 and 28) and ulcerative lesions healed to a non-ulcerative swelling in 3 animals (animals No. 22, 29 and 31).
The statement in lines 156-157 was moved to the third paragraph of discussion.
-Figure 1. Could the author enhance the quality of the bar referred to in subject n. 3 (30/38 weeks)?
The experiment with individual No. 3 lasted longer, until week 38 p.i. Graphically, it would not look good to leave this bar too long, so we chose this form of reduction. At the reviewer's request, we have redesigned the graphical representation.
Finally, despite having ulcerative lesions and a significant parasite load in their ear lobes postmortem, some cases (e.g., id 19 and 25) were not infectious at all.
Might the authors include a brief discussion on this topic?
I understand that this could be related to normal biological variation. Anyway, do the authors believe that this is due to an uneven distribution of parasites in tissues?
In other words, does the parasitic load differ at different sites on the earlobe (for example, is it higher in the section collected for qPCR?).
The reviewer is right, this matter is so interesting that we will focus a separate study on the analysis of the distribution of leishmania at the microscale and its relation to infectiousness. In order to comment on this here, we have added several sentences on this point in the sixth paragraph of the discussion:
“It is interesting to note that some individuals were not infectious for sand flies, although they had ulcerative lesions and a significant parasite load in the pinnae post mortem. We attribute this phenomenon to the heterogeneous distribution of parasites in the pinnae, as has been reported for L. donovani in the skin of BALB/c mice (Doehl et al 2017). Our next research goal will be to study the dissemination of L. major at the microscale in their natural host, Meriones shawi.”
Reviewer 2 Report
The aim of this study was to analyze the comparative contributions of asymptomatic and symptomatic hosts to the transmission of cutaneous leishmaniasis. Therefore, Meriones shawi hosts as natural reservoirs of L. major in the northern part of Africa were infected with a natural dose of L. major obtained from the gut of infected sand flies. The authors suggest that skin manifestations appeared in 90% of the animals and that xenodiagnosis with the proven vector Phlebotomus papatasi showed transmissibility in 67% of the rodents, 45% of them were repeatedly infectious to sand flies. Interestingly in xenodiagnostic trials the results suggest that there is no significant difference in the transmissibility of animals in the asymptomatic and symptomatic periods; even more, asymptomatic animals were infectious several weeks before the appearance of skin lesions and several months after their healing. The results indicate that skin lesions are not a precondition for vector infection in CL and that asymptomatic animals constitute an essential source of L. major for infection.
The manuscript herein presented, is very interesting and ingenious and demonstrate in this model the importance of asymptomatic contribution to the spread of the disease and for modelling the epidemiology of CL caused by L. major. I congratulate the authors for their work.
However, some issues have to be commented.
- What is the rationale for using the salivary gland homogenate during the infection. The clarification of this very important concept is needed for a better understanding of the “as close as possible” natural infection emulated in this manuscript.
- Again, the rationale of using 7.000 or 14.000 parasites should be stressed in order to better understand the ground of the manuscript (please see the following articles where doeses up to million parasites are used:
- Côrtes DF, Carneiro MB, Santos LM, Souza TC, Maioli TU, Duz AL, Ramos-Jorge ML, Afonso LC, Carneiro C, Vieira LQ. Low and high-dose intradermal infection with Leishmania major and Leishmania amazonensis in C57BL/6 mice. Mem Inst Oswaldo Cruz. 2010 Sep;105(6):736-45. doi: 10.1590/s0074-02762010000600002. PMID: 20944986.
b. Kimblin N, Peters N, Debrabant A, Secundino N, Egen J, Lawyer P, Fay MP, Kamhawi S, Sacks D. Quantification of the infectious dose of Leishmania major transmitted to the skin by single sand flies. Proc Natl Acad Sci U S A. 2008 Jul 22;105(29):10125-30. doi: 10.1073/pnas.0802331105. Epub 2008 Jul 14. PMID: 18626016; PMCID: PMC2481378.
- The infection of the reservoir was done in the ear pad. However, there is some migration to the lymph-node draining and to the pads could be detected. (See table 1). It would be interesting to qualify if this is high or low and discuss the importance of this finding
- Due to the importance that asymptomatic transmission is acquiring for CL and since as mentioned by the authors, only a few studies have described the infectiousness of asymptomatic hosts with CL it would be interesting to learn the opinion of the authors about the role of asymptomatic infected humans in the transmission of the disease and how to address this question.
- For all these reasons I recommend to evaluate again the manuscript after these changes are included.
Author Response
The aim of this study was to analyze the comparative contributions of asymptomatic and symptomatic hosts to the transmission of cutaneous leishmaniasis. Therefore, Meriones shawi hosts as natural reservoirs of L. major in the northern part of Africa were infected with a natural dose of L. major obtained from the gut of infected sand flies. The authors suggest that skin manifestations appeared in 90% of the animals and that xenodiagnosis with the proven vector Phlebotomus papatasi showed transmissibility in 67% of the rodents, 45% of them were repeatedly infectious to sand flies. Interestingly in xenodiagnostic trials the results suggest that there is no significant difference in the transmissibility of animals in the asymptomatic and symptomatic periods; even more, asymptomatic animals were infectious several weeks before the appearance of skin lesions and several months after their healing. The results indicate that skin lesions are not a precondition for vector infection in CL and that asymptomatic animals constitute an essential source of L. major for infection.
The manuscript herein presented, is very interesting and ingenious and demonstrate in this model the importance of asymptomatic contribution to the spread of the disease and for modelling the epidemiology of CL caused by L. major. I congratulate the authors for their work.
We sincerely thank the reviewer for the time spent on this review and the useful comments. We greatly appreciate his positive evaluation of our work. Below are point-by-point responses to all comments. The changes in the text of the manuscript are marked in blue.
However, some issues have to be commented.
- What is the rationale for using the salivary gland homogenate during the infection. The clarification of this very important concept is needed for a better understanding of the “as close as possible” natural infection emulated in this manuscript.
We added this information to the fourth paragraph of the discussion:
“We mimicked as closely as possible the natural course of initiation of infection. Sand fly salivary gland content has been added as salivary proteins and peptides with antihemostatic and immunomodulatory properties significantly affect the establishment of the parasite in the host (reviewed by Kamhawi 2000, Lestinova et al 2017).”
- Again, the rationale of using 7.000 or 14.000 parasites should be stressed in order to better understand the ground of the manuscript (please see the following articles where doeses up to million parasites are used:
- Côrtes DF, Carneiro MB, Santos LM, Souza TC, Maioli TU, Duz AL, Ramos-Jorge ML, Afonso LC, Carneiro C, Vieira LQ. Low and high-dose intradermal infection with Leishmania major and Leishmania amazonensis in C57BL/6 mice. Mem Inst Oswaldo Cruz. 2010 Sep;105(6):736-45. doi: 10.1590/s0074-02762010000600002. PMID: 20944986.
- Kimblin N, Peters N, Debrabant A, Secundino N, Egen J, Lawyer P, Fay MP, Kamhawi S, Sacks D. Quantification of the infectious dose of Leishmania major transmitted to the skin by single sand flies. Proc Natl Acad Sci U S A. 2008 Jul 22;105(29):10125-30. doi: 10.1073/pnas.0802331105. Epub 2008 Jul 14. PMID: 18626016; PMCID: PMC2481378.
We explained more our model and discussed the references in the fourth paragraph of the discussion:
“Promastigotes were taken from thoracic midguts of infected sand flies: female sand flies were dissected 8 days post bloodmeal, when mature L. major infection developed with a predominance of metacyclic stages. Infection dose was kept low, corresponding to parasite numbers in one or two infected thoracic midguts (7,000 or 14,000 parasites, respectively). Intradermal needle inoculation allowed us to control the infectious dose which is not possible with direct infection by sand fly bite, where different females inoculate the host with Leishmania numbers varying by several orders of magnitude [28-29]. Kimblin et al (2008) compared L. major infections initiated with low and high infection dose and showed that low infection dose leads to less pathology but higher parasite number in the chronic phase, so that the host act as an effective long-term reservoir of infection. With high dose (106) of parasites in inoculum, the growth of skin lesions would be unnaturally rapid as was described not only for L major, but also for L. amazonensis in mice model (Belkaid et al 2000, Cortes et al 2010).”
- The infection of the reservoir was done in the ear pad. However, there is some migration to the lymph-node draining and to the pads could be detected. (See table 1). It would be interesting to qualify if this is high or low and discuss the importance of this finding
We added a short discussion on this topic - it is now the fifth paragraph of the discussion:.
“In all animals in our experiments, parasites were most abundant in the inoculated pinnae, in two-thirds of the individuals they also spread to other areas of the body. Smaller numbers were detected in the collateral ear, the skin of the paws and tail, as well as in the lymph nodes, spleen and liver. This pattern of L. major dissemination with higher representation in the skin than in internal organs was also observed in other wild rodent species – Arvicanthis niloticus, A. neumanni, Mastomys natalensis, Cricetulus griseus and Lagurus lagurus, while extensive visceralization under the same experimental conditions was observed in Phodopus sungorus, with 71% of spleen samples infected (Vojtková et al 2020, Sadlova et al 2020).”
- Due to the importance that asymptomatic transmission is acquiring for CL and since as mentioned by the authors, only a few studies have described the infectiousness of asymptomatic hosts with CL it would be interesting to learn the opinion of the authors about the role of asymptomatic infected humans in the transmission of the disease and how to address this question.
We added a reflection on this topic to the end of discussion:
“These results also raise the question of the extent to which asymptomatic incidental hosts, including infected humans, may be involved in the transmission of CL. Only xenodiagnostic experiments covering both symptomatic and asymptomatic phases of infection can answer this question.”
- For all these reasons I recommend to evaluate again the manuscript after these changes are included.
Thank you again for all the suggestions that enriched the discussion and increased quality of our article.
Round 2
Reviewer 1 Report
Thank you very much for this second draft. The authors addressed the required comments and suggestions and now the manuscript is suitable to be published. I think that results of this impressive and labor-intensive research significantly contribute to the epidemiology of cutaneous leishmaniasis as well as they lay the basis for elaborating transmission models and control of the disease.
Best regards,